# Reason Analysis of the Jiwenco Glacial Lake Outburst Flood (GLOF) and Potential Hazard on the Qinghai-Tibetan Plateau

**Shijin Wang** [1,2] , **Yuande Yang** [3,*], **Wenyu Gong** [4] , **Yanjun Che** [1] , **Xinggang Ma** [1] **and Jia Xie** [1]

1   Yulong Snow Mountain Glacier and Environment Observation and Research Station/State Key Laboratory of Cryospheric Sciences, Northwest Institute of Eco-Environment and Resources, Chinese Academy of Sciences, Lanzhou 730000, China; wangshijin@lzb.ac.cn (S.W.); cheyanjun@jxycu.edu.cn (Y.C.); maxingang@nieer.ac.cn (X.M.); xj15193149296@163.com (J.X.)
2   College of Resources and Environment, University of Chinese Academy of Sciences, Beijing 100049, China
3   Chinese Antarctic Center of Surveying and Mapping, Wuhan University, Wuhan 430079, China
4   Institute of Geology, China Earthquake Administration, Beijing 100029, China; gwenyu@ies.ac.cn
*   Correspondence: yuandeyang@whu.edu.cn; Tel.: +86-0931-4967399

**Abstract:** Glacial lake outburst flood (GLOF) is one of the major natural disasters in the Qinghai-Tibetan Plateau (QTP). On 25 June 2020, the outburst of the Jiwenco Glacial Lake (JGL) in the upper reaches of Nidu river in Jiari County of the QTP reached the downstream Niwu Township on 26 June, causing damage to many bridges, roads, houses, and other infrastructure, and disrupting telecommunications for several days. Based on radar and optical image data, the evolution of the JGL before and after the outburst was analyzed. The results showed that the area and storage capacity of the JGL were 0.58 square kilometers and 0.071 cubic kilometers, respectively, before the outburst (29 May), and only 0.26 square kilometers and 0.017 cubic kilometers remained after the outburst (27 July). The outburst reservoir capacity was as high as 5.4 million cubic meters. The main cause of the JGL outburst was the heavy precipitation process before outburst and the ice/snow/landslides entering the lake was the direct inducement. The outburst flood/debris flow disaster also led to many sections of the river and buildings in Niwu Township at high risk. Therefore, it is urgent to pay more attention to glacial lake outburst floods and other low-probability disasters, and early real-time engineering measures should be taken to minimize their potential impacts.

**Keywords:** glacial lake outburst floods (GLOFs); outburst risk assessment; reason analysis; Qinghai-Tibetan Plateau (QTP)





## 1. Introduction

Recent reports published in *Science* and *Nature* have addressed the hazards and risks of glaciers that melt in succession [1–9]. Rapid glacial melting has caused the formation of large numbers of new glacial lakes and the expansion of existing glacial lakes and increased the potential for glacial lake outburst floods (GLOFs) [10–13]. In recent years, the glacial lake at the end of the Rongbuk Glacier on the northern slope of Mount Everest has expanded significantly. More than 12,600 glacial lakes and 250 potentially dangerous glacial lakes (PDGLs) have been identified in the Hindu Kush–Himalaya area, Chile's Patagonia, and Peru's Cordillera Blanca [14–16]. Societies and economies of GLOF-prone regions are severely impacted, including destruction of infrastructure, disruption to communities, and loss of life [16–18]. Past GLOFs have directly caused at least: 7 deaths in Iceland, 393 deaths in the European Alps, 5745 deaths in South America, and 6300 deaths in central Asia. Peru, Nepal, and India have experienced fewer floods yet higher levels of damage [14,19–21].

Affected by the persistent heavy precipitation since late May, the Jinwuco moraine lake (30°21′18.76″N, 93°37′52.81″E) in Niwu Township, Tibet Autonomous Region broke on 25 June 2020. On the next day of the outburst, the flood peak reached the downstream Niwu Township, with the storage capacity of about 5.4 million square meters. From June 26 to 28, the whole county was mobilized for flood relief, but the disaster still caused great losses to agriculture, animal husbandry, and infrastructure in the lower reaches of Nidu River. As a result, some persons were missing and numerous buildings and infrastructures were destroyed. In the same area, there was another glacial lake (Ranzeriac Lake) outburst on 5 July 2013, causing relatively large economic losses. In the next few decades, the effects of GLOFs are likely to extend farther downstream as glaciers continue to retreat in a climate warming scenario and active earthquake zone [22].

In high mountain regions where ground data collection and field observations are obstructed by harsh weather and climatic conditions, remote sensing techniques offer flexible approaches for monitoring of glacial lakes and potential GLOFs. In this context, this study conducted a special study on the cause and potential hazard of Jiwencuo GLOF by using radar and optical image data, aiming to raise the public and the government to pay high attention to GLOF with low-probability through typical cases, and take real-time engineering measures for high-risk glacial lake as soon as possible, so as to reduce the potential impact on downstream infrastructure and personnel. This study also has a certain guiding significance for the prevention of GLOF disasters in other regions.

## 2. Study Area, Data and Methods

### 2.1. Study Area

Jiwenco glacier lake (JGL) (the centroid coordinates are 30°21′21″N, 93°37′51″E) is located in the middle section of Nianqing Tanggula in the central part of the Qinghai-Tibetan Plateau (QTP) and belongs to Jiali County, Naqu Prefecture, Tibet Autonomous Region. The JGL is one of the sources of Niduzangbo (zangbo means river in Tibetan language) upstream of Yigong Zangbo, a tributary of the lower Yarlung Zangbo River (Brahmaputra River in India) (Figure 1). The highest elevation of Nidu river basin is 6883 m, while the lowest is 3110 m, with a vertical drop of more than 3500 m. The basin belongs to the plateau subfrigid subhumid monsoon climate zone. The annual mean temperature of Jiali County is −0.9 °C, and the annual precipitation is 695.5 mm. The territory of the water system development is widely distributed, in which the main rivers are Nidu and Haqu river. Affected by topography and precipitation, glaciers are well developed and belong to the temperate glaciers, and such glaciers are more sensitive to climate change because of their high temperature and fast ice flow rate [23]. There are 174 glaciers in the Nidu River Basin alone, with a total area of 417.96 km$^2$. The county has a population of about 30,000. The economy is based on animal husbandry (mainly raising yaks and sheep), with a small amount of farming, tourism, and small-scale hydroelectric power generation.

On 25 June 2020, the outburst of the JGL in the upper reaches of Nidu river in Jiari County of the QTP reached the downstream Niwu Township on June 26, causing damage to many bridges, roads, houses, and other infrastructure, and disrupting telecommunications for several days. This outburst disaster is another major disaster after the outburst disaster of the Ranzeriaco Glacier Lake (the centroid coordinates are 30°28′07″N, 93°31′55″E) in the upper reaches of Nidu river on 15 July 2013. As a result, in Niwu Township, 238 families (1160 people) were affected, 49 houses were completely destroyed, and some farmland was destroyed, leading to one cement bridge in Jiali County being washed out, the loss is estimated to be 270 million CNY (41 million USD or 35 million EUR).

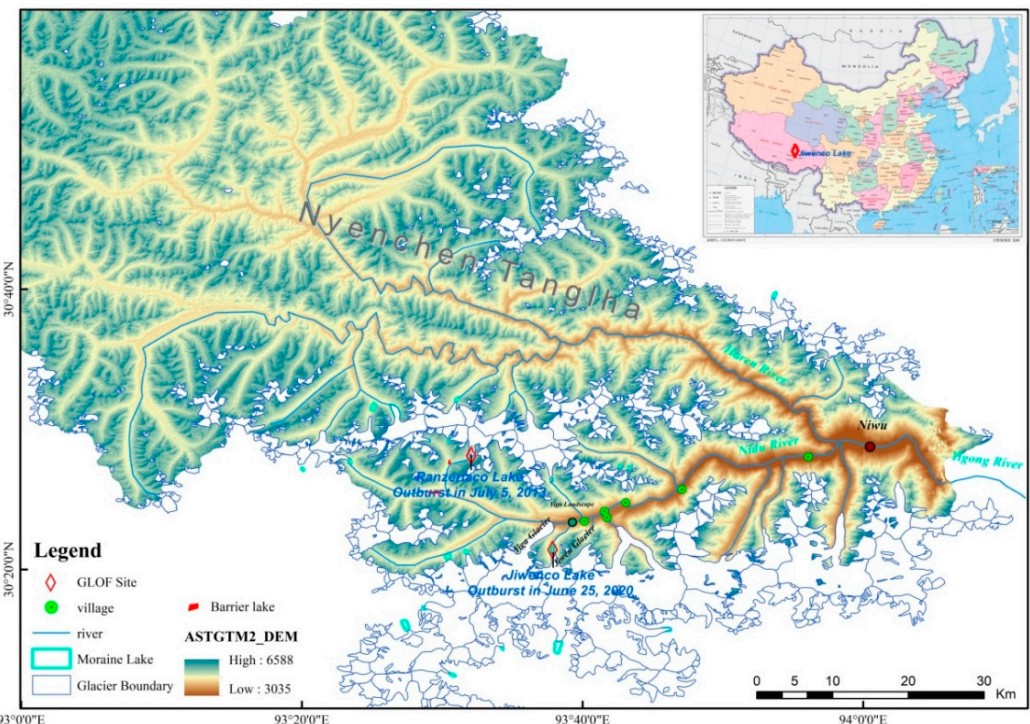

**Figure 1.** Location of Jiwenco GLOF site and Nidu river basin.

### 2.2. Data and Methods

Data used in this study include Landsat-8 OLI images, Aster remote sensing images (15 m), Sentinel-1 A, SRTM (Shuttle Radar Topography Mission) with a spatial resolution of 90 m, and Jili Meteorological Data. Specific data acquisition and methods are as follows:

### 2.2.1. Glacier Lake Area and Volume

In order to obtain the area changes before and after the outburst, three Landsat OLI images were collected by the United States Geological Survey (https://www.usgs.gov/, accessed on 4 August 2021), including Landsat OLI (30 m) (cloud cover is 3.8%) on 29 May and 27 July 2020 and 22 February 2021 before and after the lake outburst, with row numbers of 136/039. At the same time, to explore the landslide velocities and its effect on the frozen lake dam, this study obtained the Sentinel-1A images on 21 June and 3 July 2020 (https://scihub.copernicus.eu, accessed on 4 August 2021). The methods of glacier and glacial lake boundary extraction can be referred to in Guo et al. (2015) [24] and Wang et al. (2015) [22].

Satellite remote sensing data is the only accessible way to explore in detail the spatial dynamics of glaciers and glacial lake in the study area. Google Earth imageries are also utilized to validate facade variations of glacial lakes synchronized with visual interpretation.

The Annual Percentage of Area Changes (*APAC*) has been widely used to assess changes in glacier lake area in the different periods [25], and the formula is:

$$APAC = \frac{\Delta S}{S_1 \Delta t} \tag{1}$$

where *APAC* is the annual percentage of lake area change, $S_1$ is the area of the glacier lake in the before time, $\Delta S = S_2 - S_1$ is the change in area of the glacier lake during the study period, $S_2$ is the area of the glacier lake in the late time, and $\Delta t$ is the time span of the glacier lake change during the past several years. Spatial analysis tool of AcrGIS software was used to calculate the glacier lake area.

### 2.2.2. Glacier Lake Volume

Glacier lake volume was calculated by the following formula of Yao et al., 2010 [26]:

$$V = 0.0493A^{0.9304} \tag{2}$$

where, $V$ is the reservoir capacity of glacier lake in unit $km^3$, and A is the area of glacier lake in unit $km^2$.

### 2.2.3. Meteorological Dataset

Climate records of Jiali meteorological stations were used in the study. Jiali meteorological station (4488.80 m) (Code 56202 by CMDS) located in the northwest with a distance of ~47 km to Jiwen glacier has recorded the climate condition since 1961. The station was installed by China Meteorological Administration, and the data derived from the online http://data.cma.cn/ (accessed on 4 August 2021). The study focuses on the annual mean temperature and precipitation during 1961–2019, as well as the temperature and precipitation in May before the outburst in 2020. Meteorological data processing methods can be referred to Wang et al. (2013) [27].

### 2.2.4. Interpretation of Topographic Deformation

Two Sentinel-1 acquisitions, pre- (acquired on 21 June 2020) and after (acquired on 3 July 2020) the sliding event on 21 June, were used in the data processing in interpretation of topographic deformation. Sentinel-1 mission is a C band Synsetic Aperture Radar (SAR) satellite operated by European Space Agence (ESA) (https://www.esa.int/, accessed on 4 August 2021). Especially, we applied the conventional differential intermetric processes [28] to this interferometric pair. Shuttle Radar Topography Mission (SRTM) 30 m product [29] was used to remove the topography phase.

## 3. Results

### 3.1. Area Changes before and after Lake Outburst

The area and storage capacity of the JGL were 0.58 $km^2$ and 0.071 $km^3$ respectively before the outburst (on 29 May 2020), while only 0.26 $km^2$ and 0.017 $km^3$ remained after the outburst (on 27 July 2020). The outburst reservoir capacity was as high as 5.4 million $m^3$ (Figure 2).

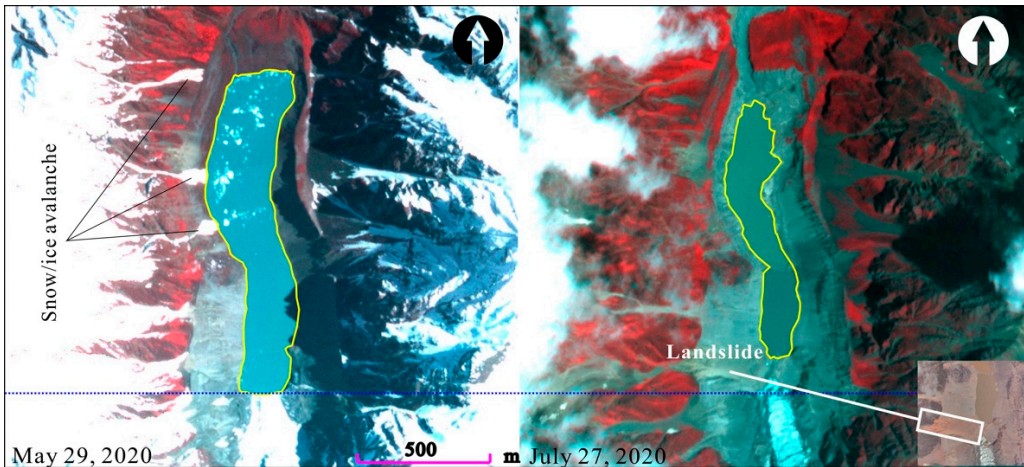

**Figure 2.** Near infrared combined image of Jinwuco glacier lake (left: before the outburst on May 29; right: after the outburst on 27 July 2020) (The blue dotted line is the end of the glacier; Avalanches or ice avalanches are the black lines on the left; the white frame is the site of landslide on the right) Image source: earthexplorer.

Before and after the outburst, the glacier connected with the glacial lake was relatively stable and changed little. On 28 February 2020, glacier length and area were 4.40 km and 9.36 km$^2$ respectively. As shown in Figure 2, the end of the glacier remained basically unchanged before and after the lake outburst, which could rule out the phenomenon of glacier surging. Therefore, it can be ruled out that the outburst was caused by glacier movement or ice avalanche in front of the glacier.

### 3.2. Loss Analysis of Outburst Disaster

After the road was repaired and opened, the research team went to the GLOF site to carry out the investigation of the JGL outburst disaster (Figure 3). It took 3 days and covered more than 100 kilometers on foot to investigate the disaster situation of the downstream glacial lake outburst flood. As of now, several buildings remain at high risk (Figure 3). According to statistics, the disaster flood washed out 0.26 km$^2$ of farmland, 14 villages about 43.9 km from the township government's house, and basically washed out roads, 6 steel bridges, 1 suspension bridge, 1 damaged cement slab culvert, 32 simple bridges, and more-than-simple houses. The total investment of 8.4 million CNY (1.30 million USD or 1.09 million EUR) in the Yiga Landscape project, 45% of which was completed, was completely submerged (Figures 1 and 3).

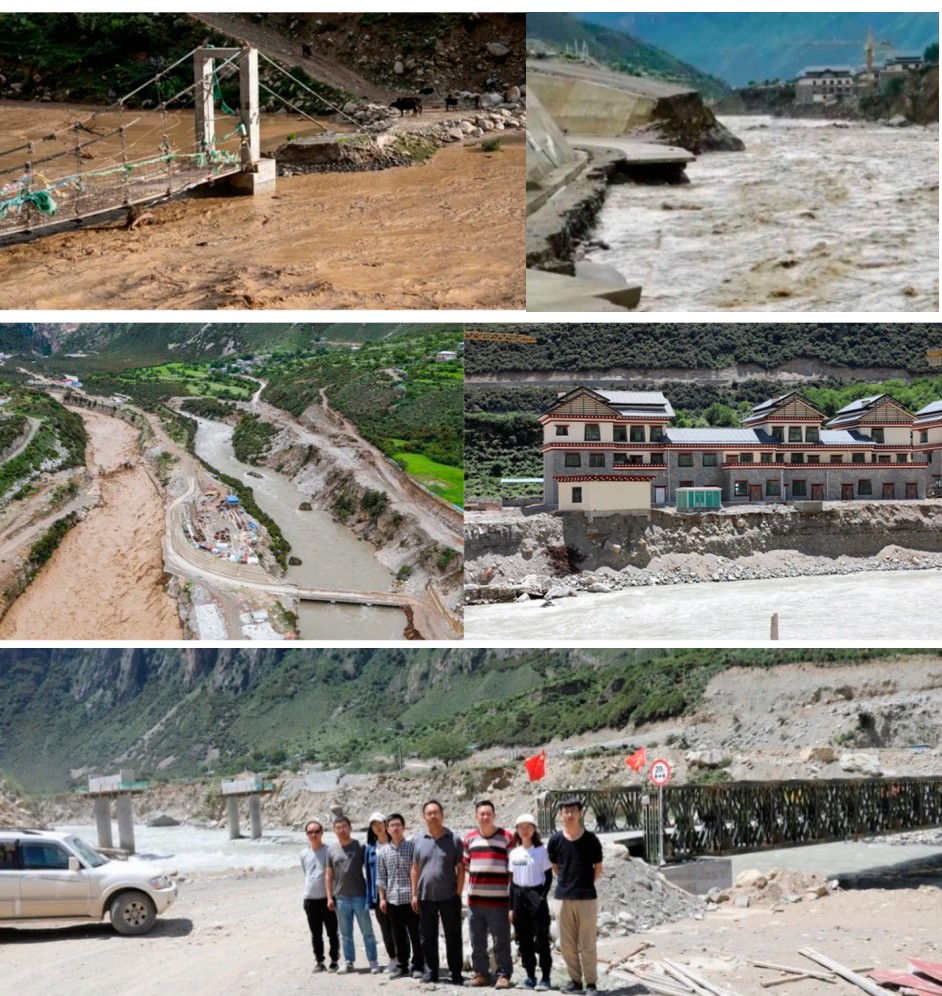

**Figure 3.** Destroyed Iron Bridge and roads and buildings still in high-risk areas (the first four photos) and restored Iron Bridge to Niwu Township (the bottom).

## 4. Discussions

### 4.1. Temperature and Precipitation Change

Glacial lake change is closely related to temperature and precipitation. Temperature changes affect the advance and retreat of glaciers, while glacial melt water and precipitation affect the amount of glacial lake water. In the last 60 years, the temperature experienced a cold period and a warm period, and an abrupt change occurred in the 1980s; the more recent the study period, the greater the temperature trend rate. The temperature showed a significant downward trend from 1961 to 1983, with a decrease rate of $-0.54$ °C/10a, while an upward trend occurred during 1984 and 2020, with a changing rate of 0.12 °C/10a, especially after 2000, with a rising rate of 0.40 °C/10a (Figure 4). From 1961 to 2020, the precipitation showed a slight increasing trend, and only increased by 23.33 mm/10a. Among them, there was a significant increasing trend from 2009 to 2020, with an increase of only 293.84 mm/10a, and the average value (777.46 mm) from 2009 to 2020 was 43.64 mm higher than the average value (733 mm) from 1961 to 2020 (Figure 4).

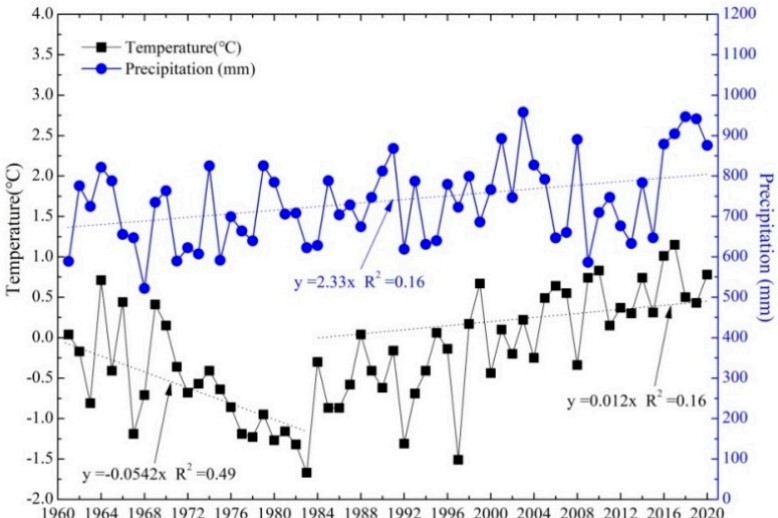

**Figure 4.** The changes of annual air temperature and precipitation of Jiali Station during 1961–2020.

Considering the altitude difference between Jiali meteorological station and JGL (4488.80 m and 4450.0 m, respectively) and the influence of mountainous terrain, the precipitation near the JGL may be more, and temperature will be lower, but the overall feature is still the same period of rain and heat. Glacial lake outburst is easily caused by intense warming and extreme precipitation [13]. According to the monthly temperature and precipitation data of Jiali Meteorological Station in June in 2020, the average monthly temperature in June was 8.28 °C, much higher than the multi-year average of 7.57 °C from 1961 to 2020. High temperatures increase the probability of avalanches, ice avalanches, and landslides, and accelerate glacier melt, which in turn increases the amount of water (ice) bodies flowing into the JGL (Figure 2). At the same time, the precipitation (240 mm) in June was far higher than the mean value of 146 mm from 1961 to 2020. Especially, there was a continuous precipitation process from May 21 to June 24 and the precipitation was as high as 224 mm (Figure 5). It can be said that the high temperature and precipitation process in June before the outburst were the main reasons for the outburst.

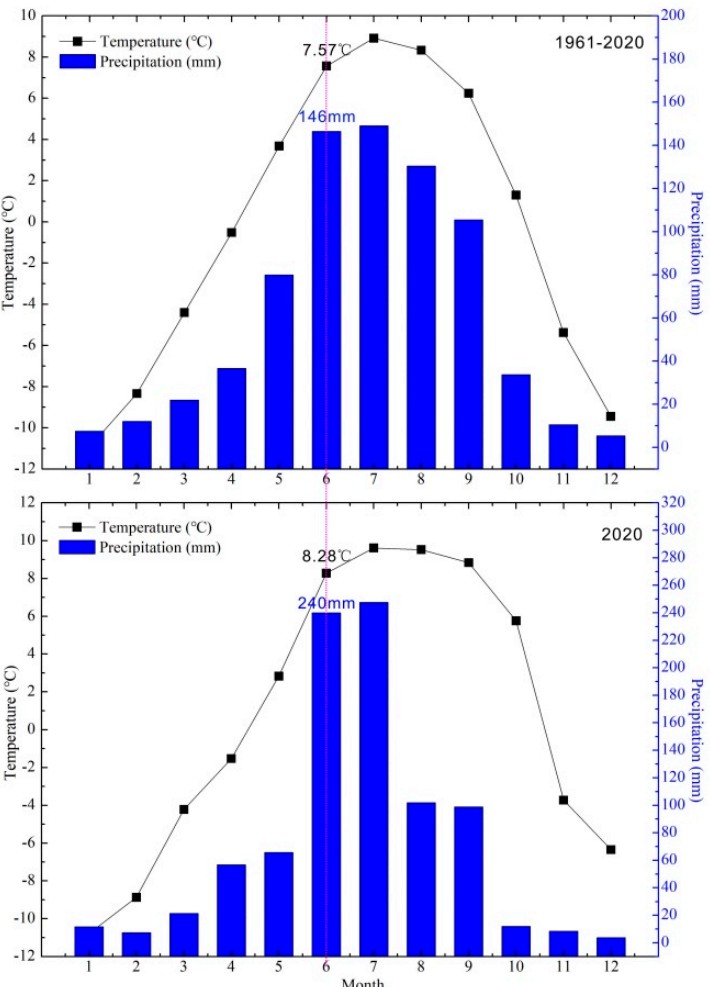

**Figure 5.** Comparison of average monthly temperature and precipitation of Jiali Station between 1961–2020 and 2020.

*4.2. Ice/snow Avalanches and Landslides*

The glacier is connected to the JGL. There is an icefall (cross-wall) between the glacier and lake, with a slope of 45° and a length of 800 m. The upper part of the ice waterfall is 4840 m above sea level, the lower part enters the lake, and the lake surface is 4450 m above sea level. The icefall and its large slope provide topographical conditions for ice avalanches. The image before the outburst on 29 May clearly showed a large amount of ice bodies distributed in the lake (Figure 2). In fact, on 5 July 2013, due to the continuous heavy rainfall and high temperature influence, ice bodies at the end of the glacier collapsed along a steep slope (cross-wall), falled glacier lake, and whipped waves topped over 30 terminal moraine dam, resulting in Ranzeriaco GLOF [30] (Figure 1). Here we generated a differential interferogram to verify result from optical remote sensing. In the end, a coherence map was obtained, as well as a wrapped interferogram as shown in Figures 6 and 7. The coherence value is an indicator of interferometric signal-to-noise ratio. In this case, the low coherence is caused by the low backscattering (e.g., water area) and rapid surface change (e.g., landslides, fast moving glacier). Figures 6 and 7 also showed that a large number of landslides entered the ice lake on 21 June, that is, an early warning could be issued on that day. After 21 June, other minor landslide events likely occurred (Greed polygon in Figure 6).

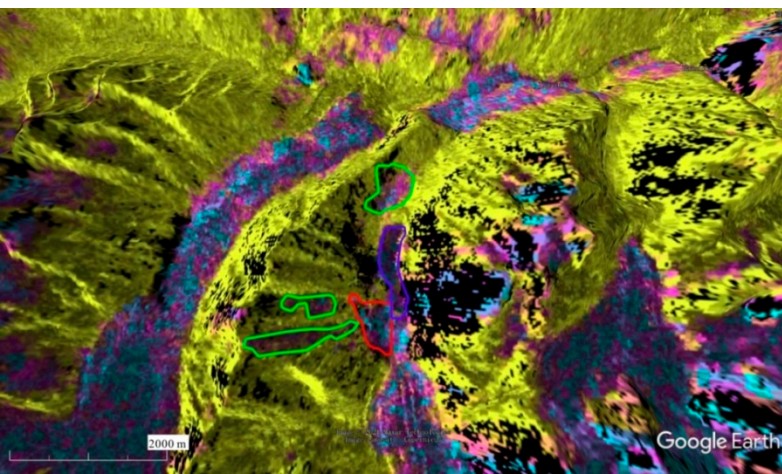

**Figure 6.** Sliding zones detected based on the filtered coherence map. The colorscale of coherence map (blue-red-yellow) indicates the coherence value from 0.1 to 0.9. The black pixels denote the shadow or layovered areas. The purple, red and green polygons denote the Jiwenco glacial lake (JGL), the collapsed landslide bodies on 21 June, and possible landslides with low coherence areas, respectively.

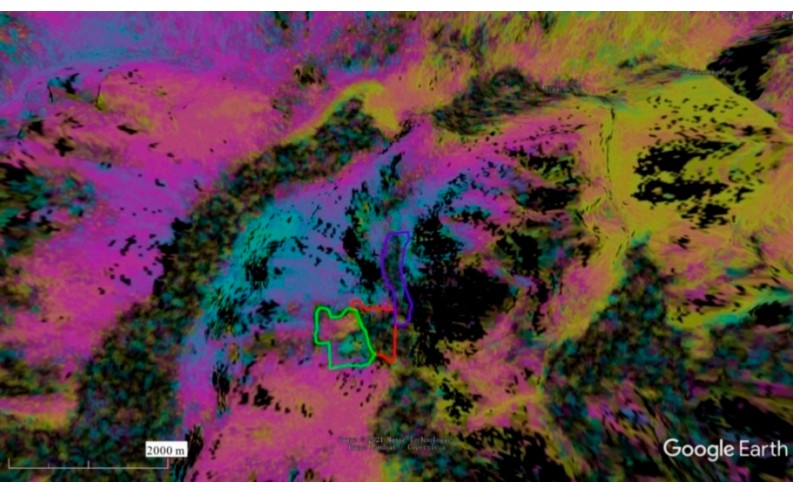

**Figure 7.** Sidling zone detected based on the wrapped interferogram. The red polygon is the same (suspected) collapsed area outlined in Figure 6. The green polygon denotes a slow sliding area above the collapsed zone.

### 4.3. Potential Risk

The two glacial lake outburst disasters in Nidu River Basin in 2013 and 2020 caused serious scouring to the riverbed and bank slope, and loose materials are widely distributed (Figure 1). The latest remote sensing image shows that the current water volume in JGL has recovered to 60% of the water volume before the outburst (2020.5.29). Meanwhile, the current Ranzeriaco lake area (0.38 km$^2$, 2018) has also recovered to 65.52% of the lake area before the outburst (0.58 km$^2$, 16 June 2013) [30]. At the same time, the Ranzeriaco outburst debris flow blocked the river channel and formed two quake lakes (0.33 km$^2$ and 0.13 km$^2$). In particular, the movement of the Yiga Glacier next to the JGL could block the Nidu River, creating a new and larger barrier lake (Figure 1). JGL, Ranzeriaco Lake, and two dammed lakes are extremely dangerous to outbursts under the background of high temperature and heavy precipitation. Many villages in Nidu river basin are located at low altitude in the valleys. The vertical drop between the JGL and the bridges and settlements is huge (the highest drop is nearly 1350 m), and the average longitudinal river slope reaches 27‰. Moreover, the Nidu river valley is relatively narrow, if the glacial lake or barrier lake has an outburst, the huge amount of water under the action of potential energy will quickly rush

to the downstream, which will cause great potential damage to Niwu Township (Figures 1 and 8).

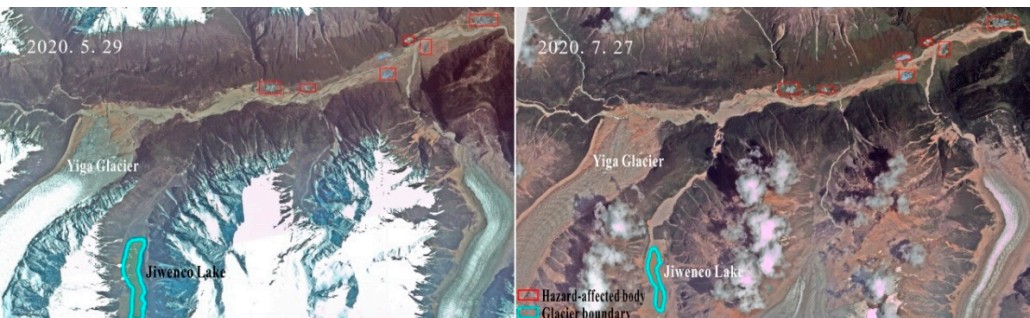

**Figure 8.** Nidu river change and distribution of hazard-affected bodies before and after the Jiwenco glacier lake outburst.

## 5. Conclusions

As it turns out, the glacial lakes fed by small glaciers cannot be ignored either. Continuous precipitation during May 2020 was the commonest cause of the JGL outburst, while high water level caused by ice/snow avalanches and landslide may be a direct cause of lake outburst. On the one hand, use of high-tech means, instruments and methods strengthens the glacial retreat, lake expansion, extreme precipitation, opening the flood flow, downstream hydrology real-time monitoring; and on the other hand, to build a frozen lake dam flood warning system, realize data information sharing, real-time for various villages, the basin area and the road department release frozen lake flood warning information. There are many glacial lakes and outburst floods that are a substantial hazard for downstream communities in the Nidu River Basin. However, unpredictable triggers and remote source locations make GLOF dynamics difficult to measure [18]. Therefore, it is urgent to use high-tech means, instruments, and methods to strengthen the comprehensive monitoring of glacier and glacial lake changes. The monitoring factors include glacier retreat, glacial lake expansion, extreme precipitation, flood discharge at the outlet, and real-time monitoring intensity of downstream hydrology. On the other hand, a glacial lake outburst flood warning system should be established to realize data and information sharing and release glacial lake flood warning information for all villages, scenic spots, and road departments in the basin in real time.

The facts indicate that the economic losses caused by GLOFs are much greater than the project costs of early consolidation of moraine dams and release of flood waters [13]. It can be said that engineering measures are the most direct way to reduce glacial lake outburst risk. The glacial lake in this area has an outburst history, so there is a large outburst probability. The Yiga Glacier landscape area under construction in the region is the path of the JGL outburst mudslide (Figure 1). In the future, the project construction and road layout should avoid the potential impact of glacial lake outburst flood and debris flow as much as possible. On the one hand, it is necessary to reinforce the outlet of the JGL, reduce the lake level or the storage capacity by siphoning measures, or build a flood spillway for reducing the risk of glacial lake outburst. At the same time, it is considered to increase the construction of debris flow, retaining the dam and drainage channel in JGL valley. In addition, it is necessary to strengthen the training of disaster prevention and mitigation for the public in the evolution area of GLOF and improve the public's awareness of mass monitoring and preventing GLOF disaster.

**Author Contributions:** Conceptualization, methodology, software and analysis, S.W.; investigation, S.W., Y.C., J.X. and X.M.; resources and data curation, Y.Y. and W.G.; writing-review and editing, S.W.; funding acquisition, S.W. All authors have read and agreed to the published version of the manuscript.

**Funding:** This research is funded by Strategic priority research program of the Chinese Academy of Sciences (XDA19070503, XDA2002010) and Applied Technology Research and Development Fund Project of Aba Prefecture, China (20YYJSYJ0076).

**Institutional Review Board Statement:** Not applicable.

**Informed Consent Statement:** Not applicable.

**Data Availability Statement:** The data presented in this study are available on request from the corresponding author.

**Acknowledgments:** We thank the TPDC for providing the first and second CGI data; NASA, NIMA, and CIAT for providing the version 4.1 SRTM data; and the CMDS for providing the meteorology data.

**Conflicts of Interest:** The authors declare no conflict of interest.

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
