# Peer review of "Reason Analysis of the Jiwenco Glacial Lake Outburst Flood (GLOF) and Potential Hazard on the Qinghai-Tibetan Plateau"

_remotesensing, doi:10.3390/rs13163114_

Round 1

Reviewer 1 Report

The paper uses a combination of meteorological measurements and several sources of remotely sensed images to evaluate a recent natural disaster in the Tibetan Plateau ( a glacial lake outburst flood event) that took place last year, causing substantial economic damage. The "reason analysis" is used to assess causality of the event in question, but the broader point seems to be that a high risk for similar events can be gleaned from this analysis, suggesting the need for mitigation strategies in the form of engineering solutions/preparedness/monitoring etc. and climate/weather trends suggest that similar events are likely in the future. The paper appears to be well written, properly organized, and scientifically sound. The topic appear to be appropriate for the journal and will likely be of interest to readers from several academic and professional disciplines. The figures appear to be of good quality and well placed. The paper appears to be well referenced/properly situated within the recent peer-reviewed literature. 

Some minor recommendations:

Line 26 -- "...Science and Nature..." -- it is appropriate to italicize these journal titles, but please do not italicize the word "and" 

Line 93 -- It might be useful here to list the financial cost of the disaster in several currencies. Many readers won't be able to intuitively evaluate the cost in Chinese currency (even though it is among the most widely traded), so it might make it easier on readers if after you list the value in RMB in parentheses, you then list the value in two or three other widely traded currencies (e.g. Euro, U.S. Dollar, Yen etc.). In this way, most readers will then be familiar with at least one of the currencies and won't have to go look up the exchange rate in order to understand the economic impact of the disaster. 

Line 124 - The formula shown here is in a much larger font than the general text. Reduce the font size to match the rest of the manuscript.  

Author Response

Line 26 -- "...Science and Nature..." -- it is appropriate to italicize these journal titles, but please do not italicize the word "and"

Italics and have been eliminated

Line 93 -- It might be useful here to list the financial cost of the disaster in several currencies. Many readers won't be able to intuitively evaluate the cost in Chinese currency (even though it is among the most widely traded), so it might make it easier on readers if after you list the value in RMB in parentheses, you then list the value in two or three other widely traded currencies (e.g. Euro, U.S. Dollar, Yen etc.). In this way, most readers will then be familiar with at least one of the currencies and won't have to go look up the exchange rate in order to understand the economic impact of the disaster.

In the Manuscript, the 2.7 million is wrong, it should be 270 million. The value of both the dollar and the euro also are increased as recommended. For example, the loss is estimated to be 270 million CNY (41 million USD or 35 million EUR).

Line 124 - The formula shown here is in a much larger font than the general text. Reduce the font size to match the rest of the manuscript.

The font size of the formula has been reduced to match the rest of the manuscript.

Reviewer 2 Report

The manuscript is interesting, well organized, though does not contain any principally new facts or ideas. Nevertheless, it is good description of new outburst of one of the Tibetian lakes.

The paper deals with glacial lake outburst flood of Jiwenco glacial lake situated in Qinghai-Tibet Plateau. The author pretends to prevent future similar disasters in other regions basing on experience of this flood. The outburst itself, its consequences and caused by them damage is presented in the paper, the rest of it is devoted to analysis of data originated from several different sources. The author describes changes of the lake area before and after the outburst, shortly analyzes the losses caused by the outburst. Then we see the analyses of air temperature, precipitation in the region for 60 years, sliding zones description. Finally, author concludes that engineering measures are the most direct way to reduce glacial lake outburst risk and it is necessary to strengthen the training of disaster prevention and mitigation for the public in evolution area of glacial lake outburst flood, and improve the public's awareness of mass monitoring and preventing of glacial lake outburst flood disaster.

There are too numerous misprints and miswrites, oscitations and neglects. E.g.: “…damage to many Bridges, roads…” (p.2, l.86), square kilometers and cubic meters are as “km2”, “m3” even in abstract, a lot of abbreviations – “GLOF”, “GLOFs” even in abstract and keywords. I’d recommend to avoid such often use of abbreviations, especially taking into account wide readers community of the journal.

Author Response

There are too numerous misprints and miswrites, oscitations and neglects. E.g.: “…damage to many Bridges, roads…” (p.2, l.86), square kilometers and cubic meters are as “km2”, “m3” even in abstract, a lot of abbreviations – “GLOF”, “GLOFs” even in abstract and keywords. I’d recommend to avoid such often use of abbreviations, especially taking into account wide readers community of the journal.

For taking into account wide readers community of the journal, numerous misprints and miswrites, oscitations and neglects have been eliminated. Similar abbreviations have been found in the relevant literature of this journal, thus only the abstract and key words have been revised, as follows:

The title: Reason analysis of the Jiwenco glacial lake outburst flood (GLOF) and potential hazard on the Qinghai-Tibetan Plateau

Abstract: Glacial lake outburst flood (GLOF) is one of the major natural disasters in the Qinghai-Tibet Plateau (QTP). On June 25, 2020, the outburst of the Jiwenco Glacial Lake (JGL) in the upper reaches of Nidu river in Jiari County of the QTP reached the downstream Niwu Township on June 26, causing damage to many bridges, roads, houses and other infrastructure and disrupting telecommunications for several days. Based on radar and optical image data, the evolution of the JGL before and after the outburst was analyzed. The results showed that the area and storage capacity of the JGL were 0.58 square kilometers and 0.071 cubic kilometers respectively before the outburst (May 29), and only 0.26 square kilometers and 0.017 cubic kilometers remained after the outburst (July 27). The outburst reservoir capacity was as high as 5.4 million cubic meters. The main cause of the JGL outburst was the heavy precipitation process before outburst and the ice/snow/landslides entering the lake was the direct inducement. The outburst flood/debris flow disaster also led to many sections of the river and buildings in Niwu Township at high risk. Therefore, it is urgent to pay more attention to glacial lake outburst floods and other low-probability disasters, and early real-time engineering measures should be taken to minimize their potential impacts.

Keywords: Glacial lake outburst floods (GLOFs); outburst risk assessment; reason analysis; Qinghai-Tibetan Plateau (QTP)